# Pharmacy Technician Efficacies and Workforce Planning: A Consensus Building Study on Expanded Pharmacy Technician Roles

**DOI:** 10.3390/pharmacy11010028

**Published:** 2023-02-03

**Authors:** Wesley Sparkmon, Marie Barnard, Meagen Rosenthal, Shane Desselle, Jordan Marie Ballou, Erin Holmes

**Affiliations:** 1Department of Pharmaceutical Sciences, Creighton University School of Pharmacy and Health Professions, Omaha, NE 68178, USA; 2Department of Pharmacy Administration, University of Mississippi School of Pharmacy, University, MS 38677, USA; 3College of Pharmacy, Touro University California, Vallejo, CA 94592, USA; 4Department of Clinical Pharmacy & Outcomes Sciences, University of South Carolina College of Pharmacy, Columbia, SC 29208, USA

**Keywords:** pharmacy technician, technician scope of practice, pharmacy workforce

## Abstract

The expansion of pharmacy technician scope of practice in recent years, though remaining somewhat contentious, has afforded multiple opportunities for pharmacy technicians to provide additional assistance within the pharmacy. However, much of the research examining this growth has focused on specific tasks, which were determined by either the researchers themselves or the respective state boards of pharmacy. This study aimed to gain a better understanding of what expanded tasks pharmacists believe technicians should have an increased role in performing. A consensus-building research methodology was used to survey practicing pharmacists to determine which tasks those pharmacists believed technicians should take an increased role in performing. This study used modified Delphi techniques to build consensus among panels of both hospital and community pharmacists regarding 20 setting-specific technician tasks. Results of our study indicated that both hospital and community pharmacists believed technicians should have an increased involvement in performing tasks which are more related to the operations of the pharmacy rather than tasks which are more clinical in nature. This finding illustrates a belief among a segment of pharmacists that expanded roles for technicians should do more to alleviate the managerial and operational burden placed on pharmacists, potentially allowing pharmacists to take on increased clinical roles.

## 1. Introduction

The expansion of pharmacy technician scope of practice within both the health system and community pharmacy settings has gained considerable traction in recent years, yet remains a contentious issue. Researchers and practitioners have examined ways to alleviate the burdens placed on pharmacists, including pharmacy technicians taking on additional responsibilities as part of their normal scope of practice. Some states have allowed technicians to complete multiple advanced tasks to aid pharmacists, while some states have limited technician role expansion. Uneven task expansion has even occurred within the same states, where regulations are applied to only certified technicians [1,2] or in different practice settings [3].

Different needs in each setting and each state means that technician scope of practice expansion has been asymmetric in its national uptake. A cursory scan of the literature finds that individual tasks have been investigated both in the United States [4,5,6,7] and internationally [8,9]. An important qualifier in many regulations surrounding technician scope of practice is the phrase “at the pharmacist’s discretion” [1]. Some states limit the abilities of the technician to the managerial discretion of pharmacists, particularly for expanded roles in the select states which have enacted such regulations [6]. Additionally, pharmacist perceptions of technicians performing certain tasks have been studied [10]. Among technicians, self-efficacy and involvement in tasks both within their normal scope of practice and beyond were measured for both settings, finding that technicians had favorable attitudes toward most activities they can be asked to perform, even those tasks technicians were generally less involved in performing [11]. However, there has been little research into which tasks pharmacists believe technicians can and should assist in performing, with many focusing on the implementation of tasks researchers believe technicians can have a role in performing [12,13,14,15,16].

Expanded technician roles were initially implemented in pilot studies prior to regulatory changes at the state level. Early studies investigated hospital technicians’ ability to accurately verify filled medication orders and found that technician verification improved efficiency in the pharmacy, decreased time to complete cart verifications, and increased patient care time for pharmacists [12,13]. Later hospital-based studies examined the benefit of including certified technicians in tasks such as medication safety procedures [14], medication reconciliation [15], and recording medication histories [16]. These expanded tasks helped pharmacists be less burdened by technical tasks and be more visible and active within patient care in many hospitals. Despite these successes, expanded technician roles and research into their applicability and utility has not expanded in the same manner in the community setting.

Researchers, however, saw the potential for expanding community pharmacy technicians’ scope of practice in the same way technician scope of practice has been expanded in hospitals. Community pharmacy pilot programs of expanded roles for technicians quickly led to regulatory changes in a small number of states that permitted technicians to provide or receive telephone prescriptions [17], administer immunizations [18,19], and verify filled prescriptions [7,20]. These regulatory changes were particularly prominent in states with larger rural populations [10], where the pharmacist might be the health care provider of closest proximity for patients.

The expansion of technician roles presents an interesting question for researchers regarding what specific tasks pharmacists believe pharmacy technicians can assist with. The regulatory differences in technician scope of practice create ambiguity but also present opportunities for researchers to investigate multiple tasks across a wide spectrum of settings and areas. Desselle et al.’s (2018) study on technician self-efficacies identified 36 tasks or activities performed by technicians in the hospital setting and 36 tasks or activities performed by technicians in the community setting [11]. This study examined both existing and emerging roles in both hospital and community pharmacy and identified factors related to pharmacy technician self-efficacies. Additionally, technicians discussed which tasks they were involved in performing currently within their normal work. Researchers have studied the effects of technicians performing some of these tasks [3,6,12,19] and the comfort of pharmacists in technicians taking on some of these additional responsibilities [10]. While these studies present evidence of success or comfort with technicians taking on additional roles such as giving or receiving verbal prescriptions over the phone [17], administering immunizations [18,19], providing non-clinical MTM services [10], or verifying prescriptions filled by other technicians [7,20], little research has examined which of these tasks do pharmacists themselves believe technicians should take an additional role in performing.

Because of the discretionary nature of advanced technician roles, it is important to increase understanding of what additional tasks pharmacists believe technicians can complete within their normal scope of practice. This all leads to the question, which tasks that most technicians are less involved in performing within the pharmacy, do pharmacists believe technicians can assist in completing? In this study, hospital and community pharmacists selected additional tasks that they consider technicians capable of performing within their normal scope of practice using consensus-building research techniques. This study aims to uncover which of these low-involvement tasks pharmacists believe that technicians have the greatest opportunity to assist in performing in their normal scope of practice.

## 2. Materials and Methods

### 2.1. Study Design

This study used modified Delphi techniques to establish which tasks pharmacists believe are most important for technicians to take on as additional roles in the pharmacy. Delphi techniques are an iterative process which are regularly used to build consensus among a panel of experts regarding a particular topic where agreement on a particular subject may not exist [21,22]. As part of this process, a panel of experts is gathered and presented a question by the researchers. Researchers synthesize the feedback from the experts and use that feedback to assist the group in reaching agreement on the studied topic. Before survey distribution, this study was exempted from review by the University of Mississippi’s Institutional Review Board (IRB).

### 2.2. Sample

A panel of 20 pharmacists (10 hospital pharmacists and 10 community pharmacists) were utilized. These pharmacists were purposively selected due to their direct interaction with technicians within their everyday practice and their ability to complete all three phases of the study and were recruited via email to participate in a three-round consensus-building study. Subjects completing all three phases of this study received a USD 50 gift card for their participation. Demographics of the sample pharmacists, including gender, education, type of practice, and state of practice are listed below in Table 1.

### 2.3. Data Collection

Participants were presented with three surveys over the course of the study. The first round of surveys was launched on 30 March 2022. The second surveys were sent to the same twenty total pharmacists on 11 April 2022, with follow-up emails sent on 18 April 2022. The third and final surveys were distributed to the sampled pharmacists on 25 April 2022, with follow-up emails sent one week later. The surveys were classified for hospital pharmacists and community pharmacists and distributed electronically using Qualtrics XM (Qualtrics, Provo, UT, USA). This method was adapted from a similar study by Bush et al. (2017) [23].

For the first distribution, each survey contained 20 tasks or activities performed by technicians from Desselle et al. (2018) [11]. Their study examined tasks which were completed by technicians along with technician self-efficacy and technician involvement in performing those tasks as part of their normal scope of practice. In our study, involvement scores were used to select surveyed tasks. In the Desselle et al. study, surveyed technicians believed they had the ability to complete the task but were not presented the opportunity to incorporate the task into their normal practice [11]. The tasks on the first survey distribution of our study had the lowest involvement scores among hospital technicians for hospital pharmacists and community technicians for community pharmacists. Tasks with highest scores remained for surveys two and three.

Pharmacists were asked to select 5 of the 20 tasks which they believe technicians should have greater involvement in within their normal scope of practice. The selection of five tasks during this phase and the second phase of the study was adapted directly from Bush et al. (2017) [23]. For each survey, space was provided for pharmacists to provide justification for their answers. Responses from the first survey distribution were recorded and frequencies were tallied so that the top 15 tasks remained for the second phase of the study. If there were not 15 tasks receiving a vote, then only tasks which were selected by a participant in the first survey were included.

The second distribution was conducted in a manner similar to the first, with pharmacists being sent the survey with the remaining tasks listed, along with one justification statement, deidentified and chosen by a member of the research team, for each task listed alongside the task. The justification statement was included to potentially persuade other pharmacists as to why a particular task would be beneficial for technicians to perform the task as part of their normal scope of practice. Pharmacists were again asked to select five of the remaining tasks and provide justification for their responses. Frequencies were again tallied, and the top 10 tasks remained for the third phase of the study. If there were not 10 tasks receiving a vote, then only tasks which were selected by a participant in the second phase were included.

For the third phase of the study, only the top 10 tasks were distributed to the panel pharmacists. Participants were asked to rank all 10 tasks based on what they believed technicians should be more involved in from most beneficial to least beneficial. The rankings were analyzed, with weighted scores being used to provide mean scores for each task.

### 2.4. Analysis

The first two phases of this study used frequencies of task selection to identify tasks to be included on the survey as part of the next phase of the study. Based on participant responses in round 3, the items were ranked according to mean scores, which were calculated according to weights for each position of an item. For example, a ranking of 1 would receive 11 points, a ranking of 2 would receive 10 points, a ranking of 3 would receive 9 points, and so on, and these scores were summed and averaged by the total number of responses.

## 3. Results

The hospital pharmacy and community pharmacy surveys were sent to 10 setting-appropriate pharmacists each. Of the 10 community pharmacists, 7 responded to the first survey distribution. The hospital survey received 11 responses despite being sent to only 10 pharmacists. Because an additional response was received, future emails clarified that the survey was not to be distributed beyond the initial 10 pharmacists. A follow-up email was sent one week after the initial distribution. After 12 days, the surveys were closed, and responses analyzed to create the second survey.

In the first community pharmacy survey, 13 of the 20 listed tasks received at least one vote and were carried through to the second survey, along with open-ended reasoning responses for remaining tasks used as supporting statements for each respective task in future surveys. The first hospital pharmacy survey saw 17 tasks receive at least one vote and 14 tasks receive at least two. Based on the criteria for number of responses to be included in the second survey of the study, 14 tasks were listed on the second survey with open-ended responses for each of the remaining tasks which were used on the second surveys as justifications for each listed task.

The second surveys were sent to the same twenty total pharmacists with the remaining tasks. As expected with multiple survey studies, response rate decreased, with only five community pharmacy responses and seven hospital pharmacy responses. After 14 days, results were analyzed to prepare the final survey distribution.

The community pharmacy survey had votes on 11 of the 13 listed tasks, with new reasoning responses listed for each selected task. These 11 selected responses and new pharmacist justification statements from the second survey distribution were listed on the final survey of the study. For the hospital pharmacy survey, all listed tasks received at least one vote, with three receiving only one. Based on our criteria, these three tasks were removed from the final survey, so only 11 tasks remained on the final survey. These tasks were listed with justification statements from the second survey.

The third and final surveys were distributed to the sampled pharmacists with the 11 remaining tasks. These surveys differed from the first two distributions, as pharmacists were instead asked to rank order the remaining 11 tasks from what they believe that technicians should be most involved in within their normal scope of practice to what pharmacists believe technicians should be least involved in performing. After 14 days, responses were analyzed and mean scores for each task among respondents was ranked from 1 to 11.

The research team received six completed surveys from the community pharmacist sample and eight completed surveys from the hospital pharmacist sample, who ranked their respective 11 tasks from first to eleventh. Responses were then scored with each first-place ranking receiving 11 points, second-place ranking receiving 10 points, and so on, with the 11th ranked tasks receiving 1 point.

Using this scoring system, results indicated that community pharmacists believed that technicians should increase involvement in supervising other technicians and in accounting and record keeping, each of which had a mean score of 7.5 among all respondents. The lowest mean scores among the remaining tasks were for discussing over-the-counter medication options with patients (mean score = 3.67) and compounding prescriptions (mean score = 4.17).

For hospital pharmacists, prescription order entry (mean score = 8) and communication with vendors and wholesalers (mean score = 7.75) were tasks which technicians should be more involved in performing as part of their normal scope of practice. Supervision of other technicians was similarly high among hospital pharmacists, with a mean score of 7.5. The lowest scores among hospital technician tasks were reconciling errors or other issues with medication administration records (mean score = 3.5) and updating medication administration records or patient’s profiles (mean score = 3.75). Table 2 and Table 3 show the total results for all 20 measured tasks across the three surveys within this study.

## 4. Discussion

Our consensus-building study found that pharmacists in both hospital and community settings were generally more in favor of technicians taking on additional tasks which assist in the operations of the pharmacy instead of tasks requiring greater clinical knowledge. When looking at the tasks which pharmacists were asked to choose from, a split emerged, particularly in later stages of the study. Despite the setting-specific differences in task involvement between hospital and community pharmacy, responses among community pharmacists (Table 3) similarly saw higher scores among more operational tasks compared with tasks requiring more clinical knowledge.

The first survey to hospital pharmacists (Table 2) illustrated that tasks which pharmacists did not believe technicians should take on included additional responsibilities of clinical communication with other healthcare professionals (“Communicate with nurses and other professionals regarding patient therapy” and “Collaborate with other health professionals to plan, monitor, review and evaluate the effectiveness of medication therapy”) or patients (“Provide information to patients on drug interactions, side effects, and medication storage”). There was also low support for technician involvement in operational tasks which had additional legal requirements, such as documentation of narcotics and staffing decisions.

In the second and third stages of the study, high scores were seen for tasks which required minimal clinical application and instead focused on more operational tasks. Pharmacists viewed the inclusion of technicians in processes such as hiring, purchasing, supervision, and transitions of care would be more beneficial to the pharmacy. These operational tasks were found to be more favorable than tasks which required some level of clinical judgment, such as verification of filled medication orders and reconciliation of errors on the medication administration records.

Another interesting finding among hospital pharmacists was that the five tasks with the highest mean scores are regularly performed within community pharmacy, but they have lower involvement scores among technicians in the institutional setting [11]. Anecdotally, pharmacy technicians in the community setting regularly participate in data entry of new prescriptions, ordering and purchasing of drug products, and billing functions. This finding somewhat illustrates the differences in job responsibilities among technicians in their respective settings.

Similar to hospital pharmacists, the panel of community pharmacists dismissed tasks in which technicians are more involved in clinical communication with other healthcare providers or patients, whether regarding medication effectiveness, lifestyle changes, or use of medical equipment. Community pharmacists also did not believe technicians should see increased involvement in tasks which increase technician responsibility in tasks where the technician would have an increased clinical or communication responsibility. The highest mean scores among community pharmacy tasks were for supervising other technicians, performing accounting and record-keeping functions, and administering immunizations. Interestingly, technician research conducted prior to the COVID-19 pandemic [10] found lower comfort levels among community pharmacists regarding technician administered immunizations. However, federal regulations during the COVID-19 pandemic permitting trained technicians to administer selected immunizations within their normal scope of practice can explain the change in opinion regarding technicians administering immunizations. 

Lower scores among community pharmacists were seen for tasks involving more advanced communications between technicians and both prescribers and patients, as well as compounding medications. Lower-scored tasks appeared to be tasks that require more clinical knowledge than the other measured tasks. Even immunization administration can be considered more operational, in that pharmacists prepare and verify the immunization before passing the immunization to the technicians to simply insert the needle and press the plunger, as it is phrased in some state rules and regulations [19]. The nature of these favored tasks of pharmacists, both hospital and community, illustrates that support for growing technician scope of practice among pharmacists is likely higher for tasks which help the operation of the pharmacy so that the pharmacist can more frequently apply their clinical knowledge [24].

As with any modified Delphi study, there are limitations to the results gathered in this study. First, the sample size of this study was low, as the total number of 20 pharmacists was divided into hospital pharmacists and community pharmacists. The small sample size is due to this study being conducted in the context of a three-part study, which can provide the framework for future data collection among pharmacists and technicians. The sample size is unlikely to be truly representative of their setting in terms of what tasks pharmacists believe technicians should be more involved in performing. However, we believe that the findings of this study can help to influence future, more expansive studies of pharmacists’ beliefs surrounding technician scope of practice. With pharmacists’ discretion being so vital to technicians taking on expanded roles [1], building knowledge of what pharmacists believe technicians should take additional responsibilities in performing is important in helping to shape technician scope of practice moving forward.

Despite the small sample size, six different states were represented within the sample. Differences in the state rules and regulations regarding the practice of pharmacy may influence responses to tasks permitted. For example, regarding technicians receiving prescriptions from prescribers, one comment said, “In the state of Mississippi, techs are [allowed] to take refills but most are not comfortable doing this. It would really save time for RPh if [a] tech can take refills on meds that are already on patient profile”. Another response stated, “[Techs can administer] COVID and flu (in Missouri). If they can administer those, they should have no trouble administering any vaccine that is [intramuscular]”. Both of these tasks had mean scores greater than or equal to 7 in survey 3, with administering immunizations being the higher of the two. It can be reasonably believed that pharmacists in states where technicians are permitted to perform more advanced tasks would like to see technician involvement in those tasks higher than pharmacists in states where technician scope of practice is more restrictive.

## 5. Conclusions

The expansion of pharmacy technician scope of practice promises new opportunities for the practice of pharmacy. States have begun finding new ways for technicians to be more involved in some of the clinical elements of pharmacy [10]. However, this study illustrated that there are tasks that technicians can take an additional role in performing without changes to the rules and regulations of many state boards of pharmacy. Pharmacists indicated that tasks which are more administrative in nature and focused on the operations of the pharmacy are tasks which technicians should take an increased role in. Changes to pharmacy policies can provide more opportunities for technicians to assist in these business functions, which can provide more clinical opportunities for the pharmacist to help their patients.

## Figures and Tables

**Table 1 pharmacy-11-00028-t001:** Demographics of sample pharmacists.

Demographic Trait		N (%)
Gender		
	Female	15 (75%)
	Male	5 (25%)
Education		
	Pharm.D.	19 (95%)
	B. Pharm	1 (5%)
Primary Practice Site		
	Community	10 (50%)
	Hospital	10 (50%)
Primary State of Practice		
	Mississippi	12 (60%)
	Kentucky	2 (10%)
	Alabama	1 (5%)
	Alaska	1 (5%)
	Georgia	1 (5%)
	Missouri	1 (5%)
	Tennessee	1 (5%)
	Texas	1 (5%)

**Table 2 pharmacy-11-00028-t002:** Results of hospital pharmacist consensus-building survey regarding technician task involvement.

Task	Round 1: Number of Votes (Total Votes = 55)	Round 2: Number of Votes (Total Votes = 35)	Round 3: Mean Scores (n = 8)	Round 3: Minimum	Round 3: Maximum	Round 3: Standard Deviation
Enter prescription orders into the computer	6	3	8	2	11	3.625
Communicate with wholesale suppliers and vendors	5	2	7.75	4	10	1.832
Supervise other technicians	6	6	7.5	2	11	3.251
Oversee activities related to medication assistance programs	3	2	7.38	2	11	3.068
Billing and other accounting functions	5	3	7	3	11	3.071
Assist with hiring other technicians	3	3	6	2	10	2.507
Assist with or facilitate patient transitions of care	4	2	6	1	10	3.891
Administer immunizations	4	4	4.63	1	9	3.335
Encourage professional development of other technicians	2	2	4.5	1	6	1.927
Update medication administration record or patient’s profile	2	3	3.75	1	7	2.053
Reconcile errors or other issues with medication administration records	3	2	3.5	1	8	2.777
Check the work of other technicians (tech-check-tech)	2	1	N/A	N/A	N/A	N/A
Preparation of clinical monitoring information for pharmacist review	2	1	N/A	N/A	N/A	N/A
Run medication utilization reports	2	1	N/A	N/A	N/A	N/A
Determine future staffing needs	1	N/A	N/A	N/A	N/A	N/A
Participate in disaster preparedness activities	1	N/A	N/A	N/A	N/A	N/A
Maintain files of narcotics and habit-forming drugs in accordance with legal requirements	1	N/A	N/A	N/A	N/A	N/A
Provide information to patients on drug interactions, side effects, and medication storage	0	N/A	N/A	N/A	N/A	N/A
Communicate with nurses and other professionals regarding patient therapy	0	N/A	N/A	N/A	N/A	N/A
Collaborate with other health professionals to plan, monitor, review, and evaluate the effectiveness of medication therapy	0	N/A	N/A	N/A	N/A	N/A

**Table 3 pharmacy-11-00028-t003:** Results of community pharmacist consensus-building survey regarding technician task involvement.

Task	Round 1: Number of Votes (Total Votes = 55)	Round 2: Number of Votes (Total Votes = 35)	Round 3: Mean Scores (n = 8)	Round 3: Minimum	Round 3: Maximum	Round 3: Standard Deviation
Enter prescription orders into the computer	6	3	8	2	11	3.625
Communicate with wholesale suppliers and vendors	5	2	7.75	4	10	1.832
Supervise other technicians	6	6	7.5	2	11	3.251
Oversee activities related to medication assistance programs	3	2	7.38	2	11	3.068
Billing and other accounting functions	5	3	7	3	11	3.071
Assist with hiring other technicians	3	3	6	2	10	2.507
Assist with or facilitate patient transitions of care	4	2	6	1	10	3.891
Administer immunizations	4	4	4.63	1	9	3.335
Encourage professional development of other technicians	2	2	4.5	1	6	1.927
Update medication administration record or patient’s profile	2	3	3.75	1	7	2.053
Reconcile errors or other issues with medication administration records	3	2	3.5	1	8	2.777
Check the work of other technicians (tech-check-tech)	2	1	N/A	N/A	N/A	N/A
Preparation of clinical monitoring information for pharmacist review	2	1	N/A	N/A	N/A	N/A
Run medication utilization reports	2	1	N/A	N/A	N/A	N/A
Determine future staffing needs	1	N/A	N/A	N/A	N/A	N/A
Participate in disaster preparedness activities	1	N/A	N/A	N/A	N/A	N/A
Maintain files of narcotics and habit-forming drugs in accordance with legal requirements	1	N/A	N/A	N/A	N/A	N/A
Provide information to patients on drug interactions, side effects, and medication storage	0	N/A	N/A	N/A	N/A	N/A
Communicate with nurses and other professionals regarding patient therapy	0	N/A	N/A	N/A	N/A	N/A
Check the work of other technicians (check-tech-check)	0	N/A	N/A	N/A	N/A	N/A

## Data Availability

The datasets generated during and/or analyzed during the current study are available from the corresponding author on reasonable request.

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
