# Peer review of "Pharmacy Technician Efficacies and Workforce Planning: A Consensus Building Study on Expanded Pharmacy Technician Roles"

_pharmacy, 2023, doi:10.3390/pharmacy11010028_

Round 1
Reviewer 1 Report
Thank you for the opportunity to review. The authors describe their study well, which aims to identify and prioritize tasks pharmacists in two practice settings feel pharmacy technicians can assist with more. To achieve this, authors used 20 tasks previously identified as low-involvement by the community and hospital-based pharmacy technicians and used a three-phase modified Delphi process to build consensus. Their introduction and discussion are well written with valid points and considerations regarding the current study and landscape.
The main concern is regarding the clarity of their methods and the confidence the authors can have in their results. The authors surveyed 20 pharmacists (10 community, 10 hospital), with varied response rates per group and phase. The description of these pharmacists, other than the practice setting, are not provided. This is significant since multiple factors may influence preferences and opinions (e.g. state of practice, experience), and it is a small sample for either group. The authors do not provide statistical analysis for the validity and certainty of their survey findings. How can readers feel confident the sample is representative of the larger population (e.g. sample size, power, or 95% confidence intervals)? The Desselle Study, for example, surveyed 5000 technicians with a 12% response rate.
Author Response
|
Reviewer 1 Comment |
Author Response |
|
The main concern is regarding the clarity of their methods and the confidence the authors can have in their results. The authors surveyed 20 pharmacists (10 community, 10 hospital), with varied response rates per group and phase. The description of these pharmacists, other than the practice setting, are not provided. This is significant since multiple factors may influence preferences and opinions (e.g. state of practice, experience), and it is a small sample for either group |
Thank you for this suggestion. We have added a demographics table to describe the sample’s gender, education, practice site, and primary state of practice. Additionally, we added context to the limitations section in our discussion regarding the small sample size of this study, namely mentioning that this study was completed as part of a larger three-part study on technician scope of practice expansion. |
|
How can readers feel confident the sample is representative of the larger population (e.g. sample size, power, or 95% confidence intervals)? |
Thank you for this comment. We did expand more context of this study to explain the small sample size of this study and what these findings could be used for in the future. |

Reviewer 2 Report
The short survey communication “Pharmacy Technician Efficacies and Workforce Planning: A Consensus Building Study on Expanded Pharmacy Technician Roles” is aimed to gain a better understanding of what expanded tasks pharmacists believe technicians should have an increased role in performing. Such a study can help to understand and expand roles for technicians should do more to alleviate the managerial and operational burden placed on pharmacists, potentially allowing pharmacists to take on the increased clinical roles.
The comments are as follows:
1. Line 52-53: However, there has been little research into which tasks pharmacists believe technicians can and should assist in performing. Can the author justify, with correlating with any literature?
2. Similar sentence is Line 87 and 88: little research has examined which of these tasks pharmacists believe technicians should take an additional role in performing. (List the task too in the introduction, that is very well discussed in previous literature and not, separately)
3. Why were pharmacists asked to select five of the 20 tasks in the first survey?
4. Line 167: Of the 10 community pharmacists, 7 responded to the first 167 surveys, while the hospital survey received 11 responses…Please rewrite
5. If literature is available compare the result with old data. How they different or similar?
6. In the discussion, if pharmacist mentioned any research or author can find any reason/hypothesis.
Author Response
|
Reviewer 2 Comment |
Author Response |
|
Such a study can help to understand and expand roles for technicians should do more to alleviate the managerial and operational burden placed on pharmacists, potentially allowing pharmacists to take on the increased clinical roles. |
Thank you. We believe this study can help lay the groundwork to build upon expanded technician roles. |
|
Thank you for this suggestion. We added additional support to this sentence by referencing the previous literature being focused on tasks selected by researchers instead of pharmacists themselves. |
|
Thank you for this comment. As with the suggestion above, specific tasks which have been previously studied in this literature were added to provide the context of what has been studied to help frame our study of what pharmacists want technicians to perform as opposed to what researchers are looking to study. |
|
We have added to our methodology section why five tasks were used (adapted from a previous study). |
|
Thank you for this recommendation. This section was rewritten for clarity. |
|
5. If literature is available compare the result with old data. How they different or similar? |
As far as previous literature, this was the first study, to our knowledge, to examine pharmacists’ opinions on tasks technicians should take an increased role in performing. Given what previous surveys have shown about technician perceived ability, a comparison would not be appropriate due to both time between and the difference in liability between pharmacists and technicians when something in the pharmacy “goes wrong”. |
|
In the discussion, context was added regarding this study being part of a larger three-part study. Because of that, there were no hypotheses regarding which tasks it was believed technicians could have an increased role in performing. |

Round 2
Reviewer 1 Report
Thank you for your response and revisions. I believe the additions will help readers make appropriate assumptions and conclusions after reading your manuscript.